# An Evaluation of the Performance of Five Burnout Screening Tools: A Multicentre Study in Anaesthesiology, Intensive Care, and Ancillary Staff

**DOI:** 10.3390/jcm10214836

**Published:** 2021-10-21

**Authors:** John Ong, Wan Yen Lim, Kinjal Doshi, Man Zhou, Ban Leong Sng, Li Hoon Tan, Sharon Ong

**Affiliations:** 1Department of Engineering, University of Cambridge, Cambridge CB2 1PZ, UK; 2Department of Medicine, National University of Singapore, Singapore 117597, Singapore; 3Department of Anaesthesiology, Sengkang General Hospital, Singhealth Services, Singapore 544886, Singapore; lim.wan.yen@singhealth.com.sg; 4Department of Anaesthesiology, Singapore General Hospital, Singhealth Services, Singapore 169608, Singapore; 5Department of Psychology, Singapore General Hospital, Singhealth Services, Singapore 169608, Singapore; kinjal.doshi@gmail.com; 6GKT School of Medical Education, King’s College, London WCR 2LS, UK; man.zhou@kcl.ac.uk; 7Department of Women’s Anaesthesia, KK Women’s and Children’s Hospital, Singhealth Services, Singapore 229899, Singapore; sng.ban.leong@singhealth.com.sg; 8Duke-NUS Medical School, Singapore 169857, Singapore; 9Department of Anaesthesia and Surgical Intensive Care, Changi General Hospital, Singhealth Services, Singapore 529889, Singapore; tan.li.hoon@singhealth.com.sg; 10Department of Surgical Intensive Care, Singapore General Hospital, Singhealth Services, Singapore 169608, Singapore

**Keywords:** burnout, burn out, stress, screening, diagnosis

## Abstract

Burnout is an important occupational hazard and early detection is paramount in preventing negative sequelae in physicians, patients, and healthcare systems. Several screening tools have been developed to replace lengthy diagnostic tools for large-scale screening, however, comprehensive head–to–head evaluation for performance and accuracy are lacking. The primary objective of this study was to compare the diagnostic performance of five burnout screening tools, including a novel rapid burnout screening tool (RBST). This was a cross-sectional study involving 493 hospital staff (anaesthesiology and intensive care doctors, nurses, and ancillary staff) at the COVID-19 frontline across four hospitals in Singapore between December 2020 and April 2021. The Maslach Burnout Inventory-Human Services Survey (MBI-HSS) was used as the reference standard. Five burnout screening tools, the single-item MBI measure of burnout (SI-MBI), dual-item MBI (DI-MBI), abbreviated MBI (aMBI), Single Item Burnout Question (SIBOQ), and the RBST, were administered via a 36-item online survey. Tools were administered simultaneously and responses were anonymised. Burnout prevalence was 19.9%. The RBST and the SI-MBI had the two highest accuracies (87.8% and 81.9% respectively) and AUROC scores (0.86, 95% CI: 0.83–0.89 and 0.86, 95% CI: 0.82–0.89 respectively). However, the accuracy of the RBST was significantly higher than the SI-MBI (*p* < 0.0001), and it had the highest positive likelihood ratio (+LR = 7.59, 95% CI 5.65–10.21). Brief screening tools detect burnout albeit with a wide range of accuracy. This can strain support services and resources. The RBST is a free screening tool that can detect burnout with a high degree of accuracy.

## 1. Introduction

Burnout is an important occupational hazard that can adversely affect patients, physicians, and healthcare systems, if unaddressed [1,2,3,4,5,6,7]. It is characterised by symptoms of emotional exhaustion, cynicism, and low professional efficacy [8,9]. The latter two are also referred to as depersonalisation or personal accomplishment [9]. Early identification of burnout is paramount in its management and allows for evidence-based individual or organisational interventions [7]. Validated tools to ‘diagnose’ burnout are invariably lengthy surveys that reduce user engagement and response [10], e.g., the 22-item Maslach Burnout Inventory-Human Services Survey (MBI-HSS), the 16-item Oldenburg Burnout Inventory (OBI), and the 19-item Copenhagen Burnout Inventory (CBI) [11,12,13]. Furthermore, the MBI-HSS, the most extensively validated and widely used tool to detect burnout in healthcare professionals [14], is proprietary and costly to administer in large populations [10], with both the survey and personal report costing up to USD 15 for an individual. This has led to the development of several burnout screening tools.

To date, most burnout screening tools have been derived from the MBI-HSS. These were created by truncating the length of the MBI-HSS survey and fielding questions with high factor loading. MBI-derived screening tools include the single-item MBI measures of burnout (SI-MBI) [15], the dual-item MBI (DI-MBI) [16], and the twelve-item abbreviated MBI (aMBI) [17]. However, the legal implications of reproducing proprietary property or parts of it without permission and suboptimal tool design remain inherent issues with these tools. Furthermore, some screening tools such as the Single Item Burnout Question (SIBOQ) [18] define abnormality using only one dimension of burnout to identify its presence. We also previously found that the aMBI was inaccurate and had a poor positive predictive value [19]. Therefore, we created a rapid burnout screening tool (RBST) that was more robust, easily administered, and reproducible without cost to our healthcare workers fronting the coronavirus disease-2019 (COVID-19) pandemic.

The primary aim of this study was to conduct a head–to–head evaluation of the accuracy of five burnout screening tools, SI-MBI, DI-MBI, aMBI, SIBOQ, and the novel RBST, validated herein. Since high levels of burnout have been reported in anaesthesiology and intensive care departments during the coronavirus disease-2019 (COVID-19) [20,21,22] pandemic, the secondary aim was to assess the prevalence of burnout and its risk factors in our frontline hospital staff within these departments.

## 2. Materials and Methods

### 2.1. Ethics

This was an anonymised and voluntary electronic survey. Respondent-identifying information was not collected. An electronic consent was completed before the start of each survey. Ethical approval was granted by the Singhealth Centralised Institutional Review Board (2020/2324).

### 2.2. Participant Selection

We hypothesised that burnout prevalence was high in COVID-19 frontline staff working in intensive care units and anaesthesiology departments, where staff are routinely involved with aerosol-generating procedures in COVID-19 patients with severe or life-threatening disease. Additionally, ancillary staff were also at risk of burnout from enforced compliance to heightened infection control measures, dynamic changes in work processes, and isolation due to social distancing measures. Our survey, therefore, included doctors and nurses from the anaesthesiology and intensive care departments and other ancillary staff (ward clerk, theatre staff, etc.). The study authors were excluded from answering the survey.

### 2.3. Design and Administration of the Survey

The 36-item electronic survey consisted of three parts. The first part comprised the 22-item MBI-HSS that assessed symptoms in the three dimensions of burnout: emotional exhaustion (EE), depersonalisation (DP), and a self-perceived lack of personal accomplishment (PA). Each response was scored on a seven-point Likert scale representing symptom frequency (ranging from 0 = never to 6 = every day). Licenses to reproduce the MBI-HSS were procured (www.mindgarden.com, accessed 5 January 2021). In setting our reference standard, we chose the MBI-HSS as it is widely regarded as the gold standard to identify burnout [23,24,25]. The SI-MBI, DI-MBI, and aMBI were derived using questions embedded within the MBI-HSS that was administered. The second part contained the RBST and SIBOQ, both of which utilised a five-point Likert scale. The third part contained multiple-choice and open-ended questions, which collected demographic and clinical data, e.g., weekly working hours (Appendix A). Respondents completed all parts in the same sitting.

The presence of EE, DP, and PA was defined using cutoff values provided in the fourth edition of the MBI manual (proprietary), with scores derived using the “average method” [11]. Briefly, cutoff values within the manual were derived from a population of 6269 healthcare workers and the following weights for each dimension were recommended as elevated EE = mean + (SD × 0.5), elevated DP = mean + (SD × 1.25), and decreased PA = mean + (SD × 0.1) [11]. The standard reference for burnout was defined as a high EE score (≥27) in combination with one or more significant dimensions, i.e., a high DP score (≥13) or a low PA score (≤31) using the MBI-HSS. These MBI criteria have been clinically validated by researchers against work-related neurasthenic symptoms listed in the World Health Organisation’s International Classification of Disease (ICD) [19,23,26,27,28]. We have previously validated these criteria in a similar population and found them reliable [19]. The proportion of respondents with single dimension EE scores ≥ 27, DP scores ≥ 10, or PA scores < 33 were also reported for academic interest.

To identify burnout, the SI-MBI utilised responses from Question 8 (Qn8) or Question (Qn10), the DI-MBI utilised responses from Qn8 and Qn10, and the aMBI utilised responses from Questions 1, 2, 5, 7, 8, 9, 10, 11, 13, 18, 19, and 20 of the MBI-HSS. For the SI-MBI, burnout was defined when symptoms of either Qn8 or Qn10 occurred at a frequency of once a week or more often, i.e., item score ≥ 4 [15]. For the DI-MBI, burnout was defined by Qn8 + Qn10 scores > 3 [16]. For the aMBI, EE, DP, and PA, scores were derived by a method we previously described [17,19]. The aMBI defined burnout as the presence of two or more ‘moderate’ scores: EE ≥ 19, DP ≥ 6, or PA ≤ 39 [17]. For the SIBOQ, burnout was defined as a positive response in either Option 3, 4, or 5 [18]. The SIBOQ has also been referred to as the single-item measure of burnout [29,30].

The RBST comprised four questions, one from each dimension of burnout: EE (Question 1), DP (Question 2), and PA (Question 3). A fourth question—“how often do you feel burnt out from your work?” provided a global assessment (Figure 1).

Questions probing respondents on how burned out they felt are featured in the MBI-HSS [11] and CBI [13]. However, the former regarded this as a subset of EE whilst the latter, a generic feature of work-related burnout. Questions chosen to feature in the RBST were based on specific themes of burnout (emotional depletion, objectification, and positive experiences) that had the highest factor loading in each MBI-HSS dimension previously determined in our population [19]. Similarly, these themes have also been reported to have one of the highest factor loadings in the MBI-HSS in other international studies [11,15,31]. The RBST was previously evaluated in a pilot study of 20 healthcare workers for reliability and validity before being administered throughout the four hospitals. In contrast to other tools, we adopted EE as a prerequisite of burnout since key opinion leaders consider EE as a cornerstone of burnout, and it possibly precedes significant symptoms of DP or PA [9,23]. Like other researchers, we considered EE significant when symptoms occurred once weekly or more frequently [15]. However, to detect the early symptoms of burnout (emerging DP or PA in the presence of severe EE symptoms), DP symptoms occurring once a month or more frequently were regarded as significant, and vice versa for PA. Finally, burnout was defined as the presence of one or more severe EE symptom(s) (marked by red squares) in combination with any significant symptom of DP or PA (marked by blue squares). This approach of using emotional exhaustion and one other significant domain to identify burnout is consistent with the clinically validated MBI criteria described above [19,23,26,27,28].

Between 20 January 2021 and 30 April 2021, survey links were disseminated via email by administrative staff within the four major hospitals that comprise the Singhealth cluster: Sengkang General Hospital (SKH), Changi General Hospital (CGH), KK Women’s and Children’s Hospital, and the Singapore General Hospital (SGH). The survey was administered in English, the working language in Singapore. Responses were collected centrally via a secure online platform. A monthly reminder email was sent during the duration of the survey.

### 2.4. Data Analyses

MedCalc V.19.1.5 was used for statistical analyses. The Cronbach’s alpha was used to determine internal consistency when three or more items were present. For better accuracy, the Spearman-Brown prophecy formula was used to determine internal consistency when screening tools contained two items only [32]. To assess construct validity, scores from dimension-specific questions in all screening tools were compared to the respective average dimension scores in the MBI-HSS (average symptom frequency of EE, DP or PA) using Spearman’s correlation since the data were nonparametric. Sensitivity, specificity, positive likelihood ratios (+LR), negative likelihood ratios (−LR), positive predictive values (PPV), negative predictive values (NPV), the area under receiver operating characteristics (AUROC) scores, and accuracy were calculated for each screening tool using true positive, true negative, false positive and false negative values, which were determined by the standard reference (MBI-HSS). Correct readouts (true-positives and true-negatives) were used to calculate tool accuracy. The McNemar test was used for head–to–head comparisons of diagnostic accuracy between burnout screening tools [33]. All *p* values reported are two-tailed and the significance level was set at 5%.

Where applicable, the Shapiro-Wilk method was used to test quantitative data for normality [34]. Subsequently, quantitative data were found to be nonparametric; therefore, summary statistics were reported as count (percentage) for categorical variables and the median and interquartile range (IQR) for continuous variables. The Chi-square test was used to compare categorical variables for univariate analysis. The Mann-Whitney U test was used to compare continuous variables between two groups. An exploratory logistic regression model was used to assess the relative contribution of different clinical factors (independent variables) to burnout status (dependent variable). Age and working hours were coded as continuous variables, whilst the remaining variables were coded as categorical variables. Only one survey form contained incomplete demographic data, and this was excluded from the logistic regression analysis.

## 3. Results

### 3.1. Reliability and Validity

In terms of reliability (internal consistency), the Cronbach’s alpha for the MBI-HSS, aMBI, and RBST were 0.8, 0.8, and 0.7 respectively. The Spearman-Brown coefficient for the SI-MBI and DI-MBI was 0.6 for both tools. The SIBOQ featured a single item; therefore, analysis for internal consistency was not applicable. For construct validity, every question within all the screening tools had statistically significant positive correlations to the MBI-related dimensions they intended to measure albeit to a varying degree (Appendix A). Criterion validity (accuracy) was assessed and reported below.

### 3.2. Sensitivity, Specificity, Predictive Values, and Likelihood Ratios

The performance statistics for all five tools are summarised in Table 1. The most sensitive tools for detecting burnout were the DI-MBI and aMBI. The lowest was the SIBOQ (80.6%). However, the DI-MBI and aMBI had the lowest specificity (64.6–66.1%), whilst the RBST had the highest specificity (89.1%). The PPVs of current screening tools were generally low allowing for our prevalence rate with the RBST being the highest at 65.4%. The NPVs were excellent in all screening tools (94.2–99.6%).

In comparing AUROC scores, the lowest score was 0.80 (SIBOQ), and the highest was 0.86 (RBST and SI-MBI). ROC curves are displayed in Figure 2A–E. Notably, the positive likelihood ratio (+LR), which is not affected by prevalence, distinguished the RBST (+LR = 7.59) from the SI-MBI (+LR = 4.48) and all other screening tools.

### 3.3. Accuracy

Considering the standard reference estimated burnout prevalence at 19.9%, all five tools that were evaluated provided population estimates to varying degrees: 25.2% (RBST), 33.0% (SIBOQ), 34.7% (SI-MBI), 48.1% (DI-MBI), and 46.9% (aMBI). Screening tools were found to have accuracies ranging from 71.4% to 87.8%. Head–to–head comparisons of the diagnostic accuracy (correct readouts out of all readouts) for each tool are displayed in Figure 2F. Briefly, the RBST was the most accurate screening tool (87.8%) and the difference between the next closest tool, the SI-MBI, was statistically significant (*p* < 0.0001). Interestingly, the SI-MBI was comparable to SIBOQ (81.9% vs. 79.1% respectively, *p* = 0.50). The least accurate were the DI-MBI and aMBI which had similar accuracies; 71.4% vs. 72.6% respectively, *p* = 0.58.

### 3.4. Demographics of Respondents

The response rate was 52.1% (493/947). 99.8% (492/493) of all respondents completed the survey in its entirety. The demographics of the respondents are summarised in Table 2. There were more female than male respondents (77.7% vs. 22.3%). However, this reflected the gender composition within the selected departments, especially among nursing staff who comprised 50.9% of our respondents. Among doctors, 45.0% were board-certified specialists, 27.2% were residents, and 27.8% were hospitalists. Among nursing staff, 12.7% were nurses-in-training, 80.9% were staff nurses, and 6.4% were clinician nurses. Doctors reported longer working hours than nurses and ancillary staff; this difference was statistically significant (*p* < 0.0001).

### 3.5. Symptoms of Burnout and Its Prevalence

Symptoms of burnout were common. According to our reference standard (MBI-HSS), approximately 1 in 5 respondents had elevated EE and DP scores. In total, 54.6% of respondents had decreased PA scores, but decreased PA scores alone is a weak predictor of burnout [9]. Burnout prevalence was 19.9% (98/493). According to job roles, burnout was present in 19.5% of doctors, 21.1% of nurses, and 16.4% of ancillary staff. Logistic regression of the variables analysed demonstrated positive associations between younger age (OR = 1.87, 95% CI: 1.35–2.65, *p* < 0.003) and those who reported an increased workload due to the pandemic (OR = 2.34, 95% CI:1.21–4.68, *p* = 0.013). Job roles and working hours did not have a statistically significant association with burnout risk (Table 3).

## 4. Discussion

This study found that the MBI-HSS, aMBI, and RBST had acceptable reliability. However, when the MBI-HSS was severely truncated to two questions, the internal consistency decreased significantly as expected. Nonetheless, all burnout screening tools demonstrated a high degree of validity. The sensitivity for detecting burnout was highest with the DI-MBI and aMBI (99.0%). However, their specificity was low, contributing to the PPVs of 41.0% and 42.0% respectively, accounting for the burnout prevalence of 19.9%. The PPVs of the SI-MBI and SIBOQ were better at 52.7% and 48.5% respectively. However, these were still suboptimal. Clinically, this implies that for every ten individuals identified as “burned out” by current screening tools, only 4 to 5 individuals in our population are actually burned out. The obligated investigation and management of the high number of “false positive” individuals would inevitably strain resources in any healthcare system. In contrast, the RBST had a higher PPV with approximately two-thirds of those identified as burned out being actually burned out. In populations with a higher burnout prevalence than ours, e.g., U.S. gastroenterologists [35], the PPV of the RBST increases accordingly and thus can provide greater benefit.

We also assessed AUROC scores to identify a screening tool with the best tradeoffs between sensitivity and specificity and found that both the RBST and SI-MBI had the highest values of 0.86. To further discern superiority between the RBST and SI-MBI, we analysed the accuracy of the screening tools for statistically significant differences. We found the RBST to be more accurate than the SI-MBI; (87.8% vs. 81.9% respectively, McNemar test *p* < 0.0001). These findings imply that the RBST can help streamline distressed individuals to appropriate channels more efficiently in our region. However, it should be emphasized that the RBST should not be used for “diagnostic” purposes as it was designed as a screening tool. A suggested approach would be to screen high-risk populations with the RBST and investigate positive results with the MBI-HSS and/or interviews.

The stark differences observed in the screening tools’ performance statistics within this study are in part due to tool design. The World Health Organisation has recently recognised burnout as an “occupational phenomenon” in the 11th International Classification of disease, acknowledging the contributions of EE, DP, and PA in syndrome characterisation. However, it has fallen short of recommending diagnostic tools or specific criteria for the syndrome [8]. Nonetheless, burnout is not characterised by a single dimension alone and screening tools that utilise single items with high factor loading in one dimension of the MBI-HSS to identify burnout can yield inaccurate results as observed in this study (i.e., a tool with high sensitivity but poor specificity). Such monodimensional approaches have been criticised as “new wine in an old bottle” by key opinion leaders such as Maslach [23]. The RBST incorporates all three dimensions of burnout, which likely explains why it fared better than other screening tools.

We acknowledge that the RBST requires further validation in different and broader populations for it to be used globally, and we are planning a study in UK gastroenterologists [36]. Nonetheless, the results are supported by a good representation of our target population (52.1% response rate), and we anticipate similar tool performance in the UK because our healthcare infrastructure was previously modelled after the UK. Similarly, we have also demonstrated that the COVID-19 pandemic has caused significant stress to our frontline healthcare workers (66.1%). Most reported an increase in workload (64.0%) and were found to be twice as likely to be burned out. Interestingly, younger age was identified as the only other independent predictor of burnout. This has also been reported as a risk factor for burnout in other studies and specialties (e.g., gastroenterology [37]). However, the reasons are poorly understood and likely multifactorial. Interestingly, the burnout prevalence rate of 19.9% in this study was found to be similar to prepandemic rates of 20.7–22.4% detected in a similar cohort of doctors [19]. The lack of a detectable increase in burnout may be explained by support measures implemented in our organisations in the early phases of the pandemic to identify and address burnout or nonparticipation of doctors who were already burned out.

This study had several limitations. Firstly, as burnout is closely associated with professional disengagement [38], nonresponse bias may have underestimated burnout prevalence. Secondly, burnout exists as a continuum and non-burned-out individuals may develop clinical burnout after an initial “negative” screen with the RBST. Thirdly, further assessment of the RBST is needed to assess its performance in different healthcare settings; we welcome this because of the significant potential of the screening tool; however, minor adjustments to the RBST may be required for different populations.

## 5. Conclusions

Burnout is common in frontline healthcare workers in the midst of the COVID-19 pandemic and brief screening tools can aid in its detection albeit with a wide range of accuracy. High rates of false positives can strain support services and resources. The RBST is a free screening tool that can detect burnout with a high degree of accuracy.

## Figures and Tables

**Figure 1 jcm-10-04836-f001:**
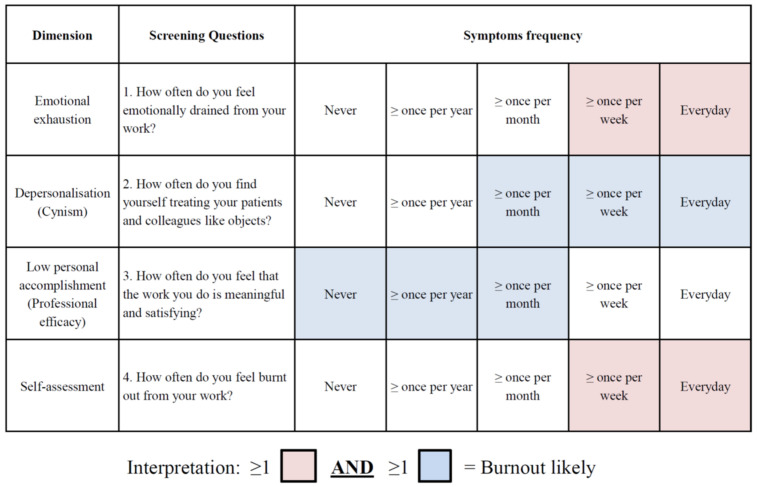
Rapid Burnout Screening Tool (RBST). A minimum of one red box AND one blue box is suggestive of burnout. Boxes in only one colour are not indicative of burnout. Note: colours provided for ease of interpretation and should not be used during administration to avoid prompting of respondents. Alternatively, coding symptom frequency may be used.

**Figure 2 jcm-10-04836-f002:**
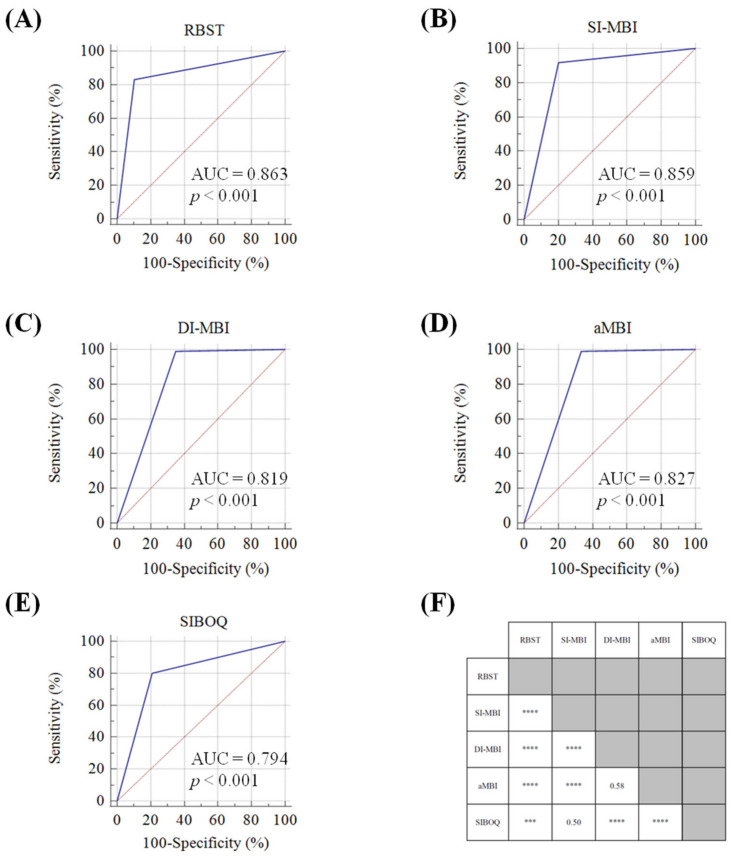
(**A**–**E**). ROC curves and comparisons of tool accuracy. ROC curves for the (**A**) Rapid Burnout Screening Tool (RBST), (**B**) Single-Item MBI measures of burnout (SI-MBI), (**C**) Dual-item MBI measure of burnout (DI-MBI), (**D**) 12-item abbreviated Maslach Burnout Inventory (aMBI), and (**E**) The Single Item Burnout Question (SIBOQ). (**F**) *p* values for head–to–head comparisons of accuracies by McNemar’s test for the five burnout screening tools. *** = *p* < 0.001, and **** = *p* < 0.0001.

**Table 1 jcm-10-04836-t001:** Performance statistics of burnout screening tools.

	RBST (4-Item)	SI-MBI (Qn8 or Qn10)	DI-MBI (Qn8 + Qn10)	aMBI (12-Item)	SIBOQ (Single Item)
Sensitivity (95% CI)	82.7% (73.7–89.6%)	91.8% (84.5–96.4%)	99.0% (94.4–100%)	99.0% (94.4–100%)	80.6% (71.4–87.9%)
Specificity (95% CI)	89.1% (85.6–92.0%)	79.4% (75.2–83.4%)	64.6% (59.6–69.3%)	66.1% (61.2–70.7%)	78.7% (74.4–82.7%)
PPV (95% CI)	65.4% (58.4–71.7%)	52.7% (47.6–57.7%)	41.0% (37.8–44.3%)	42.0% (38.7–45.4%)	48.5% (43.2–53.8%)
NPV (95% CI)	95.4% (93.1–97.0%)	97.5% (95.3–98.7%)	99.6% (97.3–99.9%)	99.6% (97.4–99.9%)	94.2% (91.6–96.1%)
+LR (95% CI)	7.59 (5.65–10.21)	4.48 (3.66–5.49)	2.79 (2.44–3.20)	2.92 (2.54–3.35)	3.79 (3.06–4.69)
−LR (95% CI)	0.20 (0.13–0.30)	0.10 (0.05–0.20)	0.02 (0.00–0.11)	0.02 (0.00–0.11)	0.25 (0.16–0.37)
AUROC (95% CI)	0.86 (0.83–0.89)	0.86 (0.82–0.89)	0.82 (0.78–0.85)	0.83 (0.79–0.86)	0.80 (0.76–0.83)
Accuracy	87.8%	81.9%	71.4%	72.6%	79.1%

CI = confidence interval, PPV = positive predictive value, NPV = negative predictive value, +LR = Positive likelihood ratio, −LR = negative likelihood ratio, AUROC = area under the receiver operating characteristics.

**Table 2 jcm-10-04836-t002:** Characteristics of respondents in the study.

Demographics	Proportion of Respondents
Gender	
Male	22.3% (110/493)
Female	77.7% (383/493)
Age groups	
20 yrs–29 yrs	27.8% (137/493)
30 yrs–39 yrs	49.1% (242/493)
40 yrs–49 yrs	15.6% (77/493)
50 yrs–59 yrs	4.9% (24/493)
≥60 yrs	2.6% (13/493)
Roles	
Doctors	34.3% (169/493)
Nurses	50.9% (251/493)
Ancillary staff	14.8% (73/493)
Weekly working hours	
Doctors, median [IQR]	55 h [45 h–65 h]
Nurses, median [IQR]	40 h [40 h–45 h]
Ancillary staff. median [IQR]	45 h [42 h–45 h]
Cared for suspected or confirmed COVID-19 patients	
Yes	47.8% (235/492)
No	52.2% (257/492)
Changes in workload secondary to pandemic	
Increased workload	64.0% (315/492)
Reduced workload	5.1% (25/492)
No change to usual workload	30.9% (152/492)
Significant increase in stress related to pandemic	
Yes	66.1% (325/492)
No	33.9% (393/492)
Burnout Prevalence	19.9% (98/493)
Emotional Exhaustion (EE)	
Summated EE score ≥ 27	26.0% (128/493)
Elevated average EE scores defined by MBI manual	26.0% (128/493)
Depersonalization (DP)/Cynicism	
Summated DP score ≥ 10	21.1% (104/493)
Elevated average DP scores defined by MBI manual	11.4% (56/493)
Low personal accomplishment (PA)/Professional efficacy	
Summated PA score < 33	54.6% (269/493)
Decreased average PA scores defined by MBI manual	71.2% (351/493)

IQR = interquartile range, COVID-19 = Coronavirus disease-2019, MBI = Maslach Burnout Inventory.

**Table 3 jcm-10-04836-t003:** Results from an exploratory logistic regression model to identify risk factors for burnout.

Variable	OR	95% CI	*p* Values
Age	1.87	1.35–2.65	0.003
Effect of the pandemic on workload			
(reference category = no change)			
Workload increased	2.34	1.21–4.68	0.013
Workload decreased	1.00	0.21–3.44	0.997
Working hours (weekly)	1.01	1.00–1.35	0.062
Increased stress caused by the Pandemic (Yes)	0.77	0.42–1.42	0.395
Gender (Male)	0.76	0.38–1.45	0.412
Hospital (Reference category = SKH)			
CGH	0.78	0.33–1.82	0.564
KKH	0.96	0.37–2.43	0.926
SGH	1.72	0.82–3.70	0.154
Caring for patients suspected with or confirmed COVID-19 (Yes)			
	0.92	0.56–1.51	0.751
Role (Reference category = Ancillary staff)			
Doctors	1.12	0.39–3.40	0.834
Nurses	1.19	0.49–2.95	0.699

SKH = Sengkang General Hospital, CGH = Changi General Hospital, KKH = KK Women’s and Children’s Hospital, SGH = Singapore General Hospital, COVID-19 = Coronavirus disease-2019.

## Data Availability

All data generated or analysed during this study are included in this published article [and its Appendix A].

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
