# Peer review of "An Evaluation of the Performance of Five Burnout Screening Tools: A Multicentre Study in Anaesthesiology, Intensive Care, and Ancillary Staff"

_jcm, 2021, doi:10.3390/jcm10214836_

Round 1

Reviewer 1 Report

Introduction line 43 - I would not use the word abnormal as it might be unhelpful or alarming. Symptoms of burnout is enough.

Throughout the manuscript it would be better to refer to 'elevated levels' of burnout or 'significant levels ' of burnout rather than abnormal - this is because the word abnormal can be stigmatising when in fact burnout is a normal response to an abnormal situation or context

The article is concise and reads well

Author Response

Dear Editor,

We thank Reviewer 1 for the constructive feedback which has enabled us to improve our manuscript. As advised, we have now performed a revision of our manuscript. Below, we address the points raised by Reviewer 1 and we have made changes to our manuscript where possible or appropriate. All page and line numbers cited are in reference to the newly uploaded manuscript with tracked changes displayed.

Reviewer 1.1              

"Introduction line 43 - I would not use the word abnormal as it might unhelpful or alarming. Symptoms of burnout is enough."

Response 1.1:            

We have deleted the word abnormal as recommended from Line 43 of the Introduction.

Reviewer 1.2:            

"Throughout the manuscript, it would be better to refer to 'elevated levels' of burnout or 'significant levels ' of burnout rather than abnormal - this is because the word abnormal can be stigmatising when in fact burnout is a normal response to an abnormal situation or context."

Response 1.2:            

Thank you. As recommended, we have now replaced the word "abnormal" with "elevated" or "decreased" when referring to burnout scores, and "significant" when referring to symptoms or dimensions of burnout throughout the manuscript.

Thank you once again for your time and feedback.

Dr Sharon Ong, on behalf of the co-authors.

Reviewer 2 Report

This research article focuses on the evaluation of screening tools for burnout.  The authors report the results from a multicentre study comparing the performance of 5 different burnout screening tools and accuracy of a newly developed tool. The topic is of interest, as burnout is indeed an important occupational hazard, often neglected. The authors have done a very good job by bringing together a very good and thorough analysis. Methods and statistical analysis performed have been presented clearly, and the new tool appears to have a good performance in screening for the presence of burnout. Therefore, I believe this paper should be accepted for publication.

I have encountered only one minor issue. In the abstract the fact that the population of interest included in the study is that of COVID-19 frontline staff is not mentioned and so is not the secondary objective of the study. Additionally, the same things have not very clearly been emphasized in the introduction.

Author Response

Dear Editor,

We thank Reviewer 2 for the constructive feedback which has enabled us to improve our manuscript. As advised, we have now performed a revision of our manuscript. Below, we address the points raised by Reviewer 2 and we have made changes to our manuscript where possible or appropriate. All page and line numbers cited are in reference to the newly uploaded manuscript with tracked changes displayed.

Reviewer 2.1              

" I have encountered only one minor issue. In the abstract, the fact that the population of interest included in the study is that of COVID-19 frontline staff is not mentioned and so is not the secondary objective of the study. Additionally, the same things have not very clearly been emphasized in the introduction."

Response 2.1:            

Thank you. In our abstract, we have now clarified that COVID-19 frontline staff were involved in the study (Page 1, Lines 30-31). We have also elaborated this in our introduction and secondary objective (Page 2, Lines 73-76).

Thank you once again for your time and feedback.

Dr Sharon Ong, on behalf of the co-authors.
